

# Integrating species and interactions into similarity metrics: a graph theory-based approach to understanding community similarity

Daniela N. López[1,2], Patricio A. Camus[3,4], Nelson Valdivia[5,6] and Sergio A. Estay[1,7]

[1] Instituto de Ciencias Ambientales y Evolutivas, Facultad de Ciencias, Universidad Austral de Chile, Valdivia, Chile
[2] Programa de Doctorado en Ciencias mención Ecología y Evolución, Facultad de Ciencias, Universidad Austral de Chile, Valdivia, Chile
[3] Departamento de Ecología, Facultad de Ciencias, Universidad Católica de la Santísima Concepción, Concepción, Chile
[4] Centro de Investigación en Biodiversidad y Ambientes Sustentables (CIBAS), Universidad Católica de la Santísima Concepción, Concepción, Chile
[5] Instituto de Ciencias Marinas y Limnológicas, Facultad de Ciencias, Universidad Austral de Chile, Valdivia, Chile
[6] Centro FONDAP de Investigación en Dinámica de Ecosistemas Marinos de Altas Latitudes (IDEAL), Universidad Austral de Chile, Valdivia, Chile
[7] Center of Applied Ecology and Sustainability (CAPES), Pontificia Universidad Católica de Chile, Santiago, Chile

Corresponding author
Daniela N. López,
danyelalopez@gmail.com

## ABSTRACT

Community similarity is often assessed through similarities in species occurrences and abundances (i.e., compositional similarity) or through the distribution of species interactions (i.e., interaction similarity). Unfortunately, the joint empirical evaluation of both is still a challenge. Here, we analyze community similarity in ecological systems in order to evaluate the extent to which indices based exclusively on species composition differ from those that incorporate species interactions. Borrowing tools from graph theory, we compared the classic Jaccard index with the graph edit distance (GED), a metric that allowed us to combine species composition and interactions. We found that similarity measures computed using only taxonomic composition could differ strongly from those that include composition and interactions. We conclude that new indices that incorporate community features beyond composition will be more robust for assessing similitude between natural systems than those purely based on species occurrences. Our results have therefore important conceptual and practical consequences for the analysis of ecological communities.

## INTRODUCTION

Characterizing the degree of similarity between ecological communities has been one of the central topics in ecology (*Devictor et al., 2010*; *Morlon, Kefi & Martinez, 2014*; *Petchey & Gaston, 2006*). In most studies, ecological similarity has mainly been evaluated by

comparing species compositions, quantifying the spatial turnover of species through cluster membership of communities and by examining the position and distance between communities in reduced species-ordination spaces (*Jaillard et al., 2018*; *Olden et al., 2004*). In this vein, β-diversity portrays the variation in species composition among spatially or temporally separated communities (*Anderson, Ellingsen & McArdle, 2006*; *Whittaker, 1972*; *Whittaker, 1960*), and thus reflects two different phenomena, spatial species turnover and nestedness of assemblages (*Baselga, 2012*; *Baselga, 2010*). β-diversity can be easily measured from species presence-absence data (*Koleff, Gaston & Lennon, 2003*; *Wilson & Shmida, 1984*) or abundances (*Barwell, Isaac & Kunin, 2015*), and both approaches include some widely studied indices such as Euclidean distances, Bray-Curtis, and Jaccard (*Anderson, Ellingsen & McArdle, 2006*). Despite the widespread use of β-diversity and other indices to characterize communities, the quantification of similarity among ecological communities, beyond species compositions remains a challenge.

In an analogous way, when ecological communities are represented by interaction networks, the similarity between spatially or temporally separated communities has been evaluated on the basis of the average number of shared nodes (species) and the degree of node overlapping (i.e., node overlapping index; *Strona & Veech, 2015*; *Strona et al., 2018*; *Zhang et al., 2016*). These measures assume that a high compositional similarity implies a high similarity in species interactions (i.e., a high "interaction similarity"). On the other hand, it has been described that species composition can be a driver of interaction turnover in mutualistic networks (*Bezemer et al., 2010*; *Trøjelsgaard et al., 2015*), but there is also evidence based on traditional methods (β-diversity) that shows a complete lack of correlation between composition similarity and interaction similarity (*Poisot et al., 2012*). Therefore, the relationship between composition and interaction similarity remains unclear.

Graph theory provides us with conceptual and practical tools that allow us to integrate the composition and interaction measurements of similarities of systems (*Ibragimov et al., 2013*; *Riesen, 2015*). These similarity measures include global metrics based on isomorphic relations or on graph transformations (*Bunke & Allermann, 1983*; *Dehmer, 2010*; *Emmert-Streib, Dehmer & Shi, 2016*; *Solé-Ribalta, Serratosa & Sanfeliu, 2012*). Similarity measures based on isomorphic relationships, quantify the exact match of nodes and links between two graphs, i.e., the "exact graph matching". Graph transformation, on the other hand, uses the concept of error-tolerant graph matching to calculate a measure of similarity based on the minimum cost (i.e., a unit of dissimilarity) of transforming one graph into another (*Dehmer, 2010*; *Riesen, 2015*). A widely used graph transformation method is the graph edit distance (GED), in which each transformation has a cost, so that a greater number of changes mirrors higher dissimilarity between the analyzed networks (*Bunke & Allermann, 1983*; *Emmert-Streib, Dehmer & Shi, 2016*). In biological sciences, GED has been used to compare protein-protein interaction (PPI) networks in human, yeast, and fruit fly, allowing researchers to infer the biological function of proteins and genes (*Ibragimov et al., 2013*; *Neyshabur et al., 2013*). By means of applying these tools to ecological data we can obtain indices that include both species composition and species interactions, providing a more complete view of community similarity.

In this study we apply conceptual tools borrowed from graph analysis to quantify community similarity, taking into account both species composition and species interactions. In addition, we compare the inferences obtained from this approach with those obtained using the classical compositional approach in order to evaluate the degree to which interaction similarity can be inferred from compositional similarity. These analyses were applied to an extensive consumer-resource interaction network.

## MATERIALS AND METHODS

### Biological data

We analyzed a dataset (https://www.nceas.ucsb.edu/interactionweb/html/thomps_towns.html) of consumer-resource interactions obtained from the National Center for Ecological Analysis and Synthesis (NCEAS). The dataset includes 16 communities (Akatore A, Berwick, Blackrock, Broad, Canton, Catlins, DempSp, German, Kyeburn, Narrowdale, NorthCol, Powder, Stony, SuttonSp, and Venlaw), covering ca. 200 km of the Taieri River tributaries in Otago, New Zealand. These sites include pine forest, broadleaf forest, pasture grassland, and tussock grassland with recorded taxonomic identities of aquatic insects, algae, and fish species and their trophic interactions (*Jaarsma et al., 1998*; *Thompson & Townsend, 2005*; *Thompson & Townsend, 2003*; *Thompson & Townsend, 2000*; *Thompson & Townsend, 1999*). The size of these networks varies from 48 to 113 taxonomic identities and from 110 to 832 consumer-resource interactions. We assign an identification code to taxonomic identities to facilitate comparison among different networks. We made some modifications to the network dataset prior to the analyses (see Data S1), checking scientific names and correcting typographical errors. Finally, similarity measures were calculated for paired combinations of sites in Taieri River (120 paired comparisons for each scenario).

### Graph edit distance (GED): concept and application

We used the graph edit distance (GED) as a metric that includes both compositional and interaction similarities. GED is a widely used graph transformation method in which each transformation (edition) necessary to pass from one network to another has a cost, so that a greater number of changes implies a higher cost, and this mirrors higher dissimilarity between the compared communities (*Bunke, 1983*; *Emmert-Streib, Dehmer & Shi, 2016*). This feature of GED represents an advantage over the composition similarity analysis, because it allows the inclusion of interaction similarity in the metrics and to assign different degrees of importance to species or interactions in the network through differential costs for each type of edition (*Bunke & Allermann, 1983*; *Emmert-Streib, Dehmer & Shi, 2016*). Despite the fact that our study only deals with trophic interactions, GED could be applied to networks containing different kinds of links, including other ecological interactions like facilitation, competition, and parasitism.

Let us consider two networks represented by the graphs $g_1 = (V_1, E_1)$ and $g_2 = (V_2, E_2)$, where V is a set of nodes, E a set of links $(u, v)$, where $u \in V$ is the source node and $v \in V$ the target node of a directed link (*Dehmer, 2010*; *Riesen, 2015*). The idea behind GED includes transforming one graph into another using edit operations $(e_i)$ such as deletions, substitutions, and insertions of nodes and links (*Bunke & Allermann, 1983*; *Dehmer, 2010*;
*Emmert-Streib, Dehmer & Shi, 2016*; *Riesen, 2015*). A given transformation is represented by an edit path ($\lambda$), which is a set of edit operations that transform $g_1$ to $g_2$. The set of all possible $\lambda$ is called $\upsilon$ ($g_1$, $g_2$) (*Riesen, 2015*). Assuming each edit operation $e_i$ has an associated cost $c(e_i)$, we can assign a relative cost to the $k^{th}$ edit path $\lambda_k$:

$$C(\lambda_k) = \sum c(e_i), \forall e_i \in \lambda_k$$

With this information we can define GED as the edit path with the minimum cost (*Dehmer, 2010*; *Ibragimov et al., 2013*),

$$GED = min[C(\lambda_k)], \forall \lambda_k \in Y(g1, g2).$$

The application of GED to the study of trophic networks allows us to evaluate differences in species compositions and those due to the absence of interactions despite species co-occurrences.

## Cost operations

Given the nature of food webs, we considered that the lowest cost was that of deleting or adding an interaction, followed by deleting or adding a species, and finally the most costly edition was that of flipping an interaction, this is a change to the hierarchy of the consumer-resource interactions (Table 1, scenarios 30, 35, 36). We considered 49 scenarios of editing costs (8,820 paired comparisons of our networks), which differed from each other in the relative magnitude of the costs of each edition operation $c(e_i)$, as shown on Table 1. We include two cost scenarios of flipping an interaction (0.25 and 0.75) represent two contrasting scenarios (low and high cost), and 5 levels of costs for deleting/adding an interaction or node. Also, eight scenarios were included specifically to assess how much GED changed by minimizing the importance of trophic interactions (reducing the costs to deleting or adding interactions, Table 1).

GEDs were calculated using the software Cytoscape (*Shannon et al., 2003*) with the GEDEVO plugin (*Ibragimov et al., 2013*), which implements an evolutionary algorithm for GED calculation. Due to GEDEVO implementing a evolutionary algorithm for estimating GED, there is no straightforward rule for stopping the iterative process. In this case we used 1,000 iterations without improvements as termination criterion for the minimization of GED. In the case of GEDEVO, GED scores ranged from 0 to 1, where 0 represented maximum similarity and 1 represented perfect dissimilarity. According to *Malek (2015)*, the method implemented in GEDEVO is that the GED of a given edit path is the result of the sum of the GED estimated for each node. The GED forming each node is transformed into a score based on the highest possible GED for a single node in the whole network.

## Compositional similarity and comparison with GED scores

In order to compare the GED results with those from a traditional compositional approach, we first computed the Jaccard index from the presence/absence species data of each network. Jaccard dissimilarity index was computed as $(a+b)/(a+b+c)$, where $a$ is the number of species only present in the first network, $b$ is the number of species only present in the second network, and $c$ is the number of species shared between both networks.

**Table 1 Scenarios of editing costs, AMI values of relationship between the similitude in species composition (1-Jaccard) and similarity of interactions (1-GED).**

| Scenarios | Flipping an interaction | Deleting or adding a species | Deleting or adding an interaction | $AMI_{(1-GED, 1-Jaccard)}$ |
|---|---|---|---|---|
| 1 | 1 | 1 | 1 | 0.032 |
| 2 | 0.25 | 0 | 0.25 | **0.134** |
| 3 | 0.25 | 0 | 0.5 | 0.071 |
| 4 | 0.25 | 0 | 0.75 | 0.064 |
| 5 | 0.25 | 0 | 1 | 0.078 |
| 6 | 0.25 | 0.25 | 0 | 0.047 |
| 7 | 0.25 | 0.25 | 0.25 | 0.074 |
| 8 | 0.25 | 0.25 | 0.5 | 0.062 |
| 9 | 0.25 | 0.25 | 0.75 | 0.073 |
| 10 | 0.25 | 0.25 | 1 | 0.065 |
| 11 | 0.25 | 0.5 | 0 | 0.039 |
| 12 | 0.25 | 0.5 | 0.25 | 0.051 |
| 13 | 0.25 | 0.5 | 0.5 | 0.035 |
| 14 | 0.25 | 0.5 | 0.75 | 0.05 |
| 15 | 0.25 | 0.5 | 1 | 0.062 |
| 16 | 0.25 | 0.75 | 0 | **0.029** |
| 17 | 0.25 | 0.75 | 0.25 | 0.047 |
| 18 | 0.25 | 0.75 | 0.5 | 0.05 |
| 19 | 0.25 | 0.75 | 0.75 | 0.037 |
| 20 | 0.25 | 0.75 | 1 | 0.05 |
| 21 | 0.25 | 1 | 0 | 0.029 |
| 22 | 0.25 | 1 | 0.25 | 0.054 |
| 23 | 0.25 | 1 | 0.5 | 0.055 |
| 24 | 0.25 | 1 | 0.75 | 0.035 |
| 25 | 0.25 | 1 | 1 | 0.041 |
| 26 | 0.75 | 0 | 0.25 | **0.084** |
| 27 | 0.75 | 0 | 0.5 | 0.051 |
| 28 | 0.75 | 0 | 0.75 | 0.024 |
| 29 | 0.75 | 0 | 1 | 0.024 |
| 30 | 0.75 | 0.25 | 0 | 0.018 |
| 31 | 0.75 | 0.25 | 0.25 | 0.074 |
| 32 | 0.75 | 0.25 | 0.5 | 0.058 |
| 33 | 0.75 | 0.25 | 0.75 | 0.044 |
| 34 | 0.75 | 0.25 | 1 | 0.031 |
| 35 | 0.75 | 0.5 | 0 | **0.014** |
| 36 | 0.75 | 0.5 | 0.25 | 0.06 |
| 37 | 0.75 | 0.5 | 0.5 | 0.05 |
| 38 | 0.75 | 0.5 | 0.75 | 0.027 |
| 39 | 0.75 | 0.5 | 1 | 0.037 |
| 40 | 0.75 | 0.75 | 0 | 0.03 |

| Scenarios | Flipping an interaction | Deleting or adding a species | Deleting or adding an interaction | $AMI_{(1-GED, 1-Jaccard)}$ |
|---|---|---|---|---|
| 41 | 0.75 | 0.75 | 0.25 | 0.047 |
| 42 | 0.75 | 0.75 | 0.5 | 0.068 |
| 43 | 0.75 | 0.75 | 0.75 | 0.039 |
| 44 | 0.75 | 0.75 | 1 | 0.068 |
| 45 | 0.75 | 1 | 0 | 0.036 |
| 46 | 0.75 | 1 | 0.25 | 0.026 |
| 47 | 0.75 | 1 | 0.5 | 0.062 |
| 48 | 0.75 | 1 | 0.75 | 0.056 |
| 49 | 0.75 | 1 | 1 | 0.04 |

In our results, the GED and Jaccard scores were expressed in terms of similarity (i.e., 1-GED and 1-Jaccard, respectively). We used the Adjusted Mutual Information metric (AMI) to assess the amount of information shared between GED and Jaccard.

Mutual Information (MI) is a measure from information theory and based on entropy estimations. MI quantifies the mutual dependence between the variables or, in other words, how much information about one variable is possible to obtain by knowing the second variable. MI can theoretically range from 0 to $\infty$. So, to standardize the values and make them comparable we used the method from *Vinh, Epps & Bailey (2010)*. In this case AMI is:

$$AMI_{max} = I(U,V) - E\{I(U,V)\} / max\{H(U), H(V)\} - E\{I(U,V)\}$$

where, $U$ and $V$ are two datasets, $I(U,V)$ represents mutual information, $E\{I(U,V)\}$ the expected mutual information, and finally $H(U,V)$ joint entropy, thus:

$$I(U,V) = \sum_{u \in U} \sum_{v \in V} p(u,v) \log p(u,v) / p(u)p(v),$$

$$H(U,V) = -\sum_{u \in U} \sum_{v \in V} p(u,v) \log p(u,v), \text{ where the entropy of a random variable is defined by:}$$

$$H(U) = -\sum p(u) \log p(u) \text{ or}$$

$$H(V) = -\sum p(v) \log p(v),$$

with probability functions $p(u)$ and $p(v)$.

The advantage of AMI is that it allows us to quantify linear and non-linear relationship between variables. AMI scores ranged from 0 to 1, where 0 represented perfect independence (no information is shared between indices) and 1 means both variables contain exactly the same information (*Vinh, Epps & Bailey, 2010*). Jaccard index and AMI calculations were performed in the R environment (*R Development Core Team, 2016*, Data S2). Jaccard index values were calculated using vegan package for R (*Oksanen et al., 2013*) and AMI values were calculated using the function discretize from the R package infotheo (*Meyer, 2014*).

We used the same approach to compare dissimilarity of interactions ($B_{WN}$) and the dissimilarity of interactions due to species turnover ($B_{ST}$) as there are defined in

**Table 2** Scenarios of editing cost, AMI values of relationship between the dissimilarity of interactions ($B_{WN}$), dissimilarity of interactions due to species turnover ($B_{ST}$) and similarity of interactions (1-GED).

| Scenarios | Flipping an interaction | Deleting or adding a species | Deleting or adding an interaction | AMI (1-GED. $B_{ST}$) | AMI (1-GED. $B_{WN}$) |
|---|---|---|---|---|---|
| 1 | 1 | 1 | 1 | ~0 | 0.017 |
| 2 | 0.25 | 0 | 0.25 | 0.014 | **0.156** |
| 3 | 0.25 | 0 | 0.5 | 0.025 | 0.099 |
| 4 | 0.25 | 0 | 0.75 | 0.027 | 0.083 |
| 5 | 0.25 | 0 | 1 | 0.007 | 0.066 |
| 6 | 0.25 | 0.25 | 0 | 0.001 | 0.034 |
| 7 | 0.25 | 0.25 | 0.25 | 0.002 | 0.059 |
| 8 | 0.25 | 0.25 | 0.5 | **0.029** | 0.102 |
| 9 | 0.25 | 0.25 | 0.75 | 0.019 | 0.086 |
| 10 | 0.25 | 0.25 | 1 | 0.023 | 0.071 |
| 11 | 0.25 | 0.5 | 0 | ~0 | 0.025 |
| 12 | 0.25 | 0.5 | 0.25 | ~0 | 0.025 |
| 13 | 0.25 | 0.5 | 0.5 | 0.004 | 0.013 |
| 14 | 0.25 | 0.5 | 0.75 | 0.002 | 0.056 |
| 15 | 0.25 | 0.5 | 1 | 0.024 | 0.05 |
| 16 | 0.25 | 0.75 | 0 | ~0 | 0.018 |
| 17 | 0.25 | 0.75 | 0.25 | ~0 | 0.019 |
| 18 | 0.25 | 0.75 | 0.5 | ~0 | 0.020 |
| 19 | 0.25 | 0.75 | 0.75 | ~0 | 0.021 |
| 20 | 0.25 | 0.75 | 1 | 0.015 | 0.049 |
| 21 | 0.25 | 1 | 0 | 0.002 | 0.021 |
| 22 | 0.25 | 1 | 0.25 | ~0 | 0.014 |
| 23 | 0.25 | 1 | 0.5 | ~0 | 0.017 |
| 24 | 0.25 | 1 | 0.75 | ~0 | 0.009 |
| 25 | 0.25 | 1 | 1 | ~0 | 0.03 |
| 26 | 0.75 | 0 | 0.25 | ~0 | 0.088 |
| 27 | 0.75 | 0 | 0.5 | 0.006 | 0.045 |
| 28 | 0.75 | 0 | 0.75 | 0.007 | 0.032 |
| 29 | 0.75 | 0 | 1 | 0.01 | 0.009 |
| 30 | 0.75 | 0.25 | 0 | 0.005 | 0.015 |
| 31 | 0.75 | 0.25 | 0.25 | 0.015 | 0.068 |
| 32 | 0.75 | 0.25 | 0.5 | 0.009 | 0.04 |
| 33 | 0.75 | 0.25 | 0.75 | ~0 | 0.025 |
| 34 | 0.75 | 0.25 | 1 | 0.007 | 0.037 |
| 35 | 0.75 | 0.5 | 0 | ~0 | 0.019 |
| 36 | 0.75 | 0.5 | 0.25 | ~0 | 0.055 |
| 37 | 0.75 | 0.5 | 0.5 | 0.006 | 0.035 |
| 38 | 0.75 | 0.5 | 0.75 | ~0 | 0.015 |
| 39 | 0.75 | 0.5 | 1 | ~0 | 0.023 |
| 40 | 0.75 | 0.75 | 0 | 0.011 | 0.029 |

**Table 2** (*continued*)

| Scenarios | Flipping an interaction | Deleting or adding a species | Deleting or adding an interaction | AMI (1-GED. $B_{ST}$) | AMI (1-GED. $B_{WN}$) |
|---|---|---|---|---|---|
| 41 | 0.75 | 0.75 | 0.25 | 0.015 | 0.031 |
| 42 | 0.75 | 0.75 | 0.5 | ~0 | 0.038 |
| 43 | 0.75 | 0.75 | 0.75 | ~0 | 0.022 |
| 44 | 0.75 | 0.75 | 1 | ~0 | 0.037 |
| 45 | 0.75 | 1 | 0 | ~0 | 0.024 |
| 46 | 0.75 | 1 | 0.25 | ~0 | 0.011 |
| 47 | 0.75 | 1 | 0.50 | ~0 | 0.032 |
| 48 | 0.75 | 1 | 0.75 | 0.006 | 0.036 |
| 49 | 0.75 | 1 | 1 | ~0 | 0.008 |

*Poisot et al. (2012)* with the estimated GED values in the 49 scenarios (Table 2). $B_{WN}$ and $B_{ST}$ values were calculated using betalink package for R (*Poisot et al., 2012*). Finally, we used the AMI to compared the amount of information shared between GED and $B_{WN}$, and between GED and $B_{ST}$. The objective of this comparison is to evaluate the relationship of GED with another recent methodological approach to the problem, and also to evaluate the relationship of GED under different cost scenarios with the result obtained by partitioning network similarity into species and interaction β-diversity as is proposed by *Canard (2011)* and *Poisot et al. (2012)*.

## RESULTS

The interaction similarity scores based on the GED ranged from 0.099 to 0.99 for the Taieri River networks. On the other hand, compositional similarity (Jaccard) ranged from 0.08 to 0.67. The highest values of GED were found when the interactions had zero cost (Fig. 1A). We found lower values of GED when the adding/deleting nodes had zero cost and adding/deleting interactions had a low cost (Figs. 1C and 1D, Table 1). A general pattern in the relationship between Jaccard and GED can be observed in Fig. S1. Through almost all scenarios, at low levels of the Jaccard index (<0.2), estimated GED values tend to be highly variable, but at moderate-high values (>0.2) the relationship turns asymptotic (Figs. 1C and 1D, Fig. S1). Due to this relationship, AMI values were low, regardless of the scenario. Estimated values were far from 1, with a maximum of 0.134 for scenario 2 and minimum of 0.014 for scenario 35 (Fig. 2, Table 1). The interpretation of these markedly low AMI values irrespective of the scenario means both metrics seem to be unrelated and do not share significant amounts of information (see Fig. 2). In practical terms, network similarity (nodes and links) cannot be inferred or predicted from similitude at the species level.

We also found a weak relationship between $B_{ST}$/$B_{WN}$ and GED across all scenarios (Fig. 3, Table 2). Despite of this, AMI values were consistently higher for the relationship between $B_{WN}$ and GED (AMI from 0.008 to 0.156) than between $B_{ST}$ and GED (AMI from ~0 to 0.029). The maximum AMI value was observed when the cost of adding/deleting nodes was lower than adding/deleting links (Table 2).
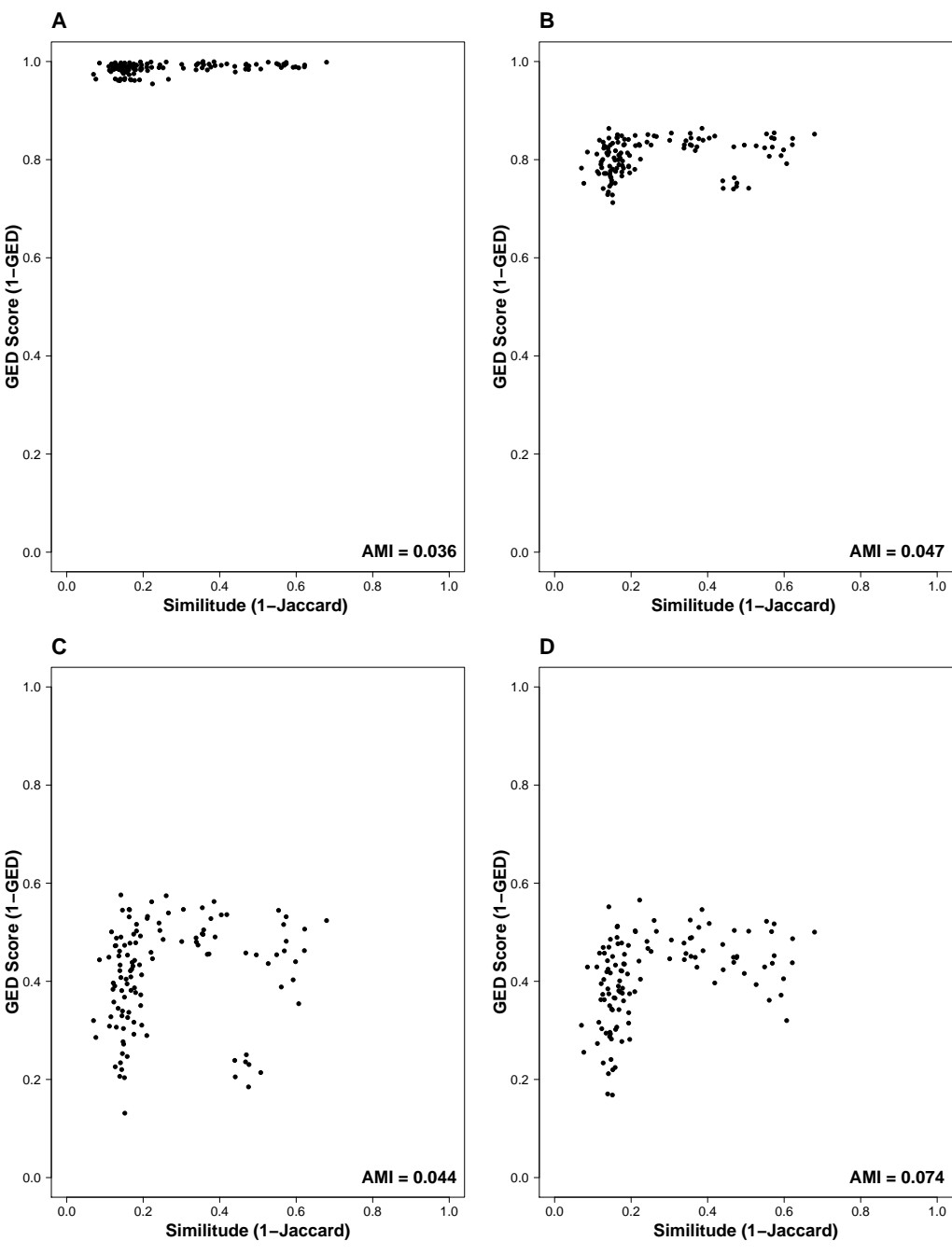

**Figure 1** **Relationship between the similitude in species composition (1-Jaccard) and similarity of interactions (1-GED) in grassland systems.** (A) Flipping: 0.75, Add or delete nodes: 1, Add or delete links: 0, (B) Flipping: 0.75, Add or delete nodes: 0.75, Add or delete links: 0.25, (C) Flipping: 0.75, Add or delete nodes: 0.25, Add or delete links: 0.75, (D) Flipping: 0.25, Add or delete nodes: 0, Add or delete links:1. AMI values are included in each plot. Each point represents a pairwise comparison between food webs.

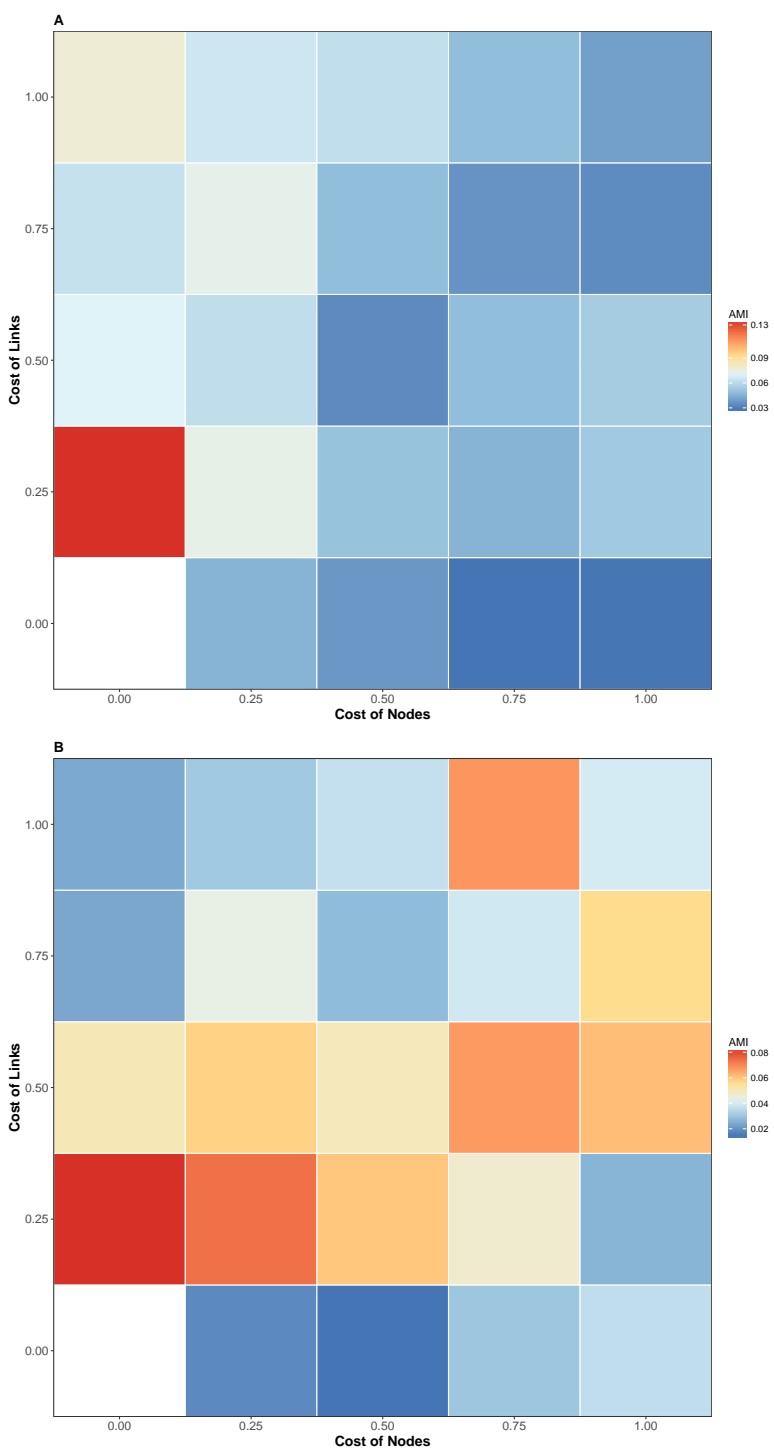

**Figure 2** **AMI values of relationship between the similitude in species composition (1-Jaccard) and similarity of interactions (1-GED) in grassland systems.** Color codes shows the AMI values. (A) Flipping 0.25, (B) Flipping 0.75.

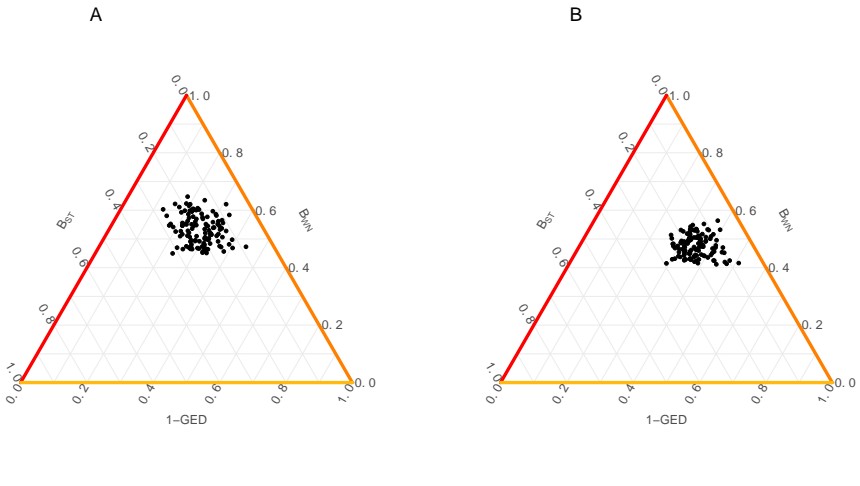

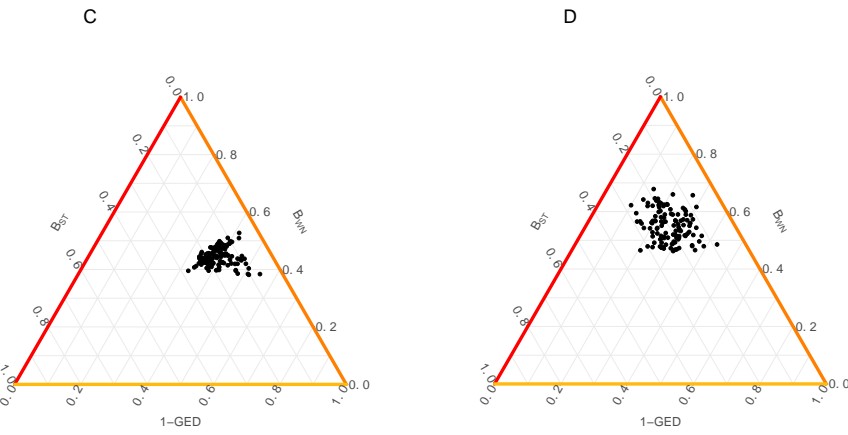

**Figure 3** **Relationship among the dissimilarity of interactions due to species turnover ($B_{ST}$), dissimilarity of interactions ($B_{WN}$) and Graph Edit Distance (1-GED) in grassland systems.** (A) Flipping: 0.75, Add or delete nodes: 1, Add or delete links: 0,25, (B) Flipping: 0.75, Add or delete nodes: 1, Add or delete links: 0.50, (C) Flipping: 0.75, Add or delete nodes: 0.25, Add or delete links: 0.25, (D) Flipping: 0.75, Add or delete nodes: 0.25, Add or delete links: 1.

## DISCUSSION

We found a weak relationship between community similarities measured using the Jaccard and GED indices; the AMI values were far from those expected for both metrics equally describing the communities. Similar results have been described by other studies in which differences between similarity metrics (composition similarity—interaction similarity) has been explained by phenomena such as changes in phenology (*Edwards & Richardson, 2004*), trophic interactions mediated by the presence of a third species, and changes in abundance, behavior, or physiology (*Burkle, Marlin & Knight, 2013*; *Poisot et al., 2012*). Undoubtedly, these phenomena confer variation to communities in different ways, and thus affect the occurrence of both species and trophic interactions. In the same vein, changes in climatic conditions can decouple the synchrony of ecological processes, such as predator–prey interactions (*Harrington, Woiwod & Sparks, 1999*; *Thackeray et al., 2010*). This implies that short-term changes in species' seasonal phenology could result in the loss of some predator–prey interactions, thus impacting interaction similarities. Likewise, mechanisms such as changes in species abundances and predator behavior can jointly affect the trophic similarity of communities (*Abrams, 1982*; *Arditi & Ginzburg, 2012*). Examples of these mechanisms include changes in abundances that affect predator–prey encounter probabilities (*Poisot, Stouffer & Gravel, 2015*), and prey behavioral changes causing prey to be less vulnerable given high predator densities (*Charnov, 1976*; *Skalski & Gilliam, 2001*). Thus, all these pieces of evidence suggest that species co-occurrence is necessary but not the only factor that influences the occurrence of ecological interactions (*Lopez et al., 2017*; *Poisot et al., 2012*; *Thompson & Townsend, 1999*).

*Canard (2011)* and *Poisot et al. (2012)* developed a similar approach based on β-diversity measure of dissimilarity, traditionally used to assess community similarity. When comparing these methods with the approach presented here, results showed that both approaches do not seem to share information (as reflected in AMI values). This could suggest that, because of their different origin, both approaches are not measuring exactly the same dimension of community similarity. However, GED values were significantly more related to $B_{WN}$ than to $B_{ST}$ despite the cost scenario. This is not surprising because, as it is calculated, $B_{WN}$ contains to $B_{ST}$, so this apparent pattern is just a consequence of $B_{ST}$ being a subset of $B_{WN}$. In this context, we think that a valuable advantage of using GED is the flexibility of assigning different degrees of importance to species or interactions in the network through differential costs for each type of edition (*Bunke & Allermann, 1983*; *Emmert-Streib, Dehmer & Shi, 2016*). This characteristic of GED allows researchers to focus the analysis into the specific components of their interest.

On the other hand, GED is a measure of the similarity between pairwise networks that incorporates both species composition and the structure of interactions, this implies that GED includes compositional metrics. In this vein, two communities with high values of Jaccard would be also similar in functionality (high GED or similar paths of energy flows) if and only if the occurrence of the species is strongly related to the occurrence of the interactions.

Recently, other authors have proposed a different approach to integrate compositional and "interactional" community similarities (*Schmidt, Rodrigues & Von Mering, 2017*). In this approach, species co-occurrences are used to infer biotic interactions, through similarity-based network inference. This kind of inference includes constructing a network from pairwise co-occurrences and/or mutual exclusion, and some metric is used to quantify and determine the significance of the similarity of the pairwise distributions (*Faust & Raes, 2012*; *Schmidt, Rodrigues & Von Mering, 2017*). Despite the fact that these methods have been successfully applied in microbial ecology, and their use has complemented the information provided by standard analytical approaches (*Barberán et al., 2012*; *Cazelles et al., 2016*; *Faust & Raes, 2012*; *Freilich et al., 2018*; *Morales-Castilla et al., 2015*; *Schmidt, Rodrigues & Von Mering, 2017*), the utility of this method to analyze interactions at macroscales is questionable, mainly due to the independence between co-occurrence measures and interactions (*Akin & Winemiller, 2006*; *Lopez et al., 2017*; *Poisot et al., 2012*; *Saavedra et al., 2016*; *Thompson & Townsend, 1999*). Some key restrictions of this theoretical framework have been associated to its inability to exactly reproduce the interaction networks (*Freilich et al., 2018*; *Morales-Castilla et al., 2015*), and that co-occurrences alone are not sufficient to provide insight about the biotic interactions in these communities. As such, several have noted the importance of directly collecting information of trophic interactions when analyzing natural communities (*Cazelles et al., 2016*; *Morales-Castilla et al., 2015*). This being said, collecting information about trophic interactions is expensive and time-consuming, so in many cases the implementation of this kind of sampling is not possible. Overall, given that metrics used to describe communities do not necessarily share significant levels of information, researchers should clearly identify not only their research question, but also the metrics (richness, composition, evenness, abundance, interactions) that best describe the dimension of the community to be analyzed.

The method used in this study reveals the effect that trophic interactions have on community similarity and highlights the constraints at analyzing community similarity in one or a few ecological dimensions. Similarly, others have developed a new approach to separate the effects of interactions in ecosystem functioning from those of species composition (*Jaillard et al., 2018*). These authors have found that the effects of interactions and composition are independent, but both contribute significantly to ecosystem functioning. Thus, the direct interpretation of species interactions from co-occurrence data remains controversial (*Cazelles et al., 2016*). In our case, and unlike microbial ecology studies, it was not necessary to infer trophic interactions because they were directly measured. Furthermore, here the AMI values confirmed that species co-occurrence and interactions did not equally portray the similarity of the communities regardless of the different cost scenarios assigned to the species and the structure of the interaction network.

The data analysis used in this study could be useful when interactions between species should be inferred from co-occurrence data. In these cases, potential bias from assumed relationships between composition and interactions could be avoided by attributing different costs when calculating GED. For example, the impact of interaction similarities on the whole similarity could be approximated by assigning different costs to link editions. If the whole similarity is notoriously affected by link edition costs then co-occurrences

and interactions are likely independent. Because the selection of a particular cost scenario could be considered subjective, and so affect the global result of the analysis, we think that following some simple criteria can reduce this subjectiveness. In our case and following the point of view of an ecologist, the deletion of a node in a network can reflect the temporal extinction of a species, and an addition reflects the opposite. The same can be applied to links that represent interactions. We considered that deleting or adding links to a network has a lower cost due to the high variability of the analyzed trophic interactions over time (*Lopez et al., 2017*), and because the presence of an interaction depends hierarchically on species composition (a given interaction occurs if and only if both interacting species co-occur). The flipping of an interaction was attributed a higher cost, because this edition changes the hierarchy of the consumer-resource interactions and the direction of energy transfer within the network, which is an ecologically costly and rare phenomenon.

In the ecological literature some level of association between state variables like richness, abundances, species composition, or biomass is usually assumed; however, many of these variables does not show robust association patterns (*Edwards & Richardson, 2004*; *Poisot et al., 2012*; *Pool et al., 2016*). The poor association between the similarity indices shown in this study reinforces the idea that each state variable represents a different dimension of natural systems. Future studies could take advantages of methods like those used in this study to ensure the robust assessment of the similitude of natural systems.

## CONCLUSIONS

In summary, our results show: (a) a weak relationship between measures of similarity using only the species composition and that those include composition and interactions, evidence the need of adding structural relationships to the similarity measure and (b) GED can be advantageous in the analysis of networks and ecological communities due to its flexibility in assigning different cost schemes depending of the researcher interest. A more practical consequence of our results is a cautionary note on community-level interpretation of similarity. In the analysis of ecological communities, it is commonly assumed that different sites, assemblages, or communities can be considered replicates. However, the question that arises is how similar must communities be in terms of any state variable to represent an adequate replicate of the system?

Finally, our results show that graph edit distance (GED) might be a valuable metric for the analysis of ecological communities. The use of an integrated measure that allows us to incorporate information of the composition and interaction structure of communities is useful for effectively establishing whether two communities, rather than operational units, could be considered as equivalent ecological systems. Our results further give warning about the need to account for the particularities of the multiple state variables that represent dimensions of ecological systems.

## ACKNOWLEDGEMENTS

We thank the Interaction Web Database of National Center for Ecological Analysis and Synthesis (https://www.nceas.ucsb.edu/interactionweb/) at the University of California, Santa Barbara, USA.

### Funding

Financial support was provided by CONICYT Grant N° 21140959 (DNL), FONDECYT N° 1040425 (PAC), 1190529 (Nelson Valdivia), 1160370 (SAE) and to the Center of Applied Ecology and Sustainability (CAPES) CONICYT PIA/BASAL FB0002 (SAE). Nelson Valdivia was supported by FONDAP IDEAL grant N° 15150003. The funders had no role in study design, data collection and analysis, decision to publish, or preparation of the manuscript.

### Grant Disclosures

The following grant information was disclosed by the authors:
CONICYT: N° 21140959.
FONDECYT: N° 1040425, 1190529, 1160370.
Center of Applied Ecology and Sustainability (CAPES) CONICYT PIA/BASAL: FB0002.
FONDAP IDEAL: N° 15150003.

### Competing Interests

The authors declare there are no competing interests.

### Author Contributions

- Daniela N. Lopez analyzed the data, contributed reagents/materials/analysis tools, prepared figures and/or tables, authored or reviewed drafts of the paper, approved the final draft.
- Patricio A. Camus and Nelson Valdivia contributed reagents/materials/analysis tools, authored or reviewed drafts of the paper, approved the final draft.
- Sergio A. Estay analyzed the data, contributed reagents/materials/analysis tools, authored or reviewed drafts of the paper, approved the final draft.

### Data Availability

The datasets are available in Dataset S1. The original data set is available at the Web Database of the National Center for Ecological Analysis and Synthesis (NCEAS), at the University of California, Santa Barbara, USA.

(https://www.nceas.ucsb.edu/interactionweb/html/thomps_towns.html).

### Supplemental Information

Supplemental information for this article can be found online at http://dx.doi.org/10.7717/peerj.7013#supplemental-information.

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
