# Peer review of "Integrating species and interactions into similarity metrics: a graph theory-based approach to understanding community similarity"

_PeerJ, doi:10.7717/peerj.7013_

## Round 0.1 · original submission · Major Revisions

The two referees have provided positive comments and constructive feedback with suggestions to improve the paper. Can you please attend to these carefully and in doing so, give the paper another careful proof read for formatting. Also make sure the figures use a minimum of colours and 'clutter' and their text will be clearly legible if reduced in the final publication.

·

Basic reporting

In this manuscript, the authors examine how the graph edit distance (GED)
could be used to evaluate similarity between ecological communities.
According to me, this is an interesting topic well described in the
introduction, the methods are readable and the findings well reported.
Moreover, the manuscript is well structured and conforms to PeerJ standards.
Figures are relevant and of good quality. I however I did not understand where
the second figure (the pdf file `nyl.pdf`) is supposed to be.

A website to retrieve raw data is mentioned and I was able to use them.
However, given what the authors explain lines 111-113, I think it would
better if the authors could provide the cleaned data set. Also, as the authors only
used one of the data set avaiable on the Interaction Web Database, I
recommend to add the following link in the manuscript:
https://www.nceas.ucsb.edu/interactionweb/html/thomps_towns.html

As a non-native English speakers, I do not feel comfortable enough to evaluate
the quality of the English language used. That said, I think it is overall
clear and unambiguous, I was however able to detect one grammatical error in the
acknowledgment section, line 274: "We thank to Interaction Web Database of".

Experimental design

## What beta-diversity diversity

It is indeed relevant to quantify how the estimations of dissimilarity based
on GED differ from the one based on a common index of similarity (here the
Jaccard index). However I believe that the authors should also
compare their method to the one developed by @poisot_dissimilarity_2012.
The authors first mention this study lines 74-75:

> [...], but there is also evidence showing a complete
lack of correlation between composition similarity and interaction similarity
(Poisot et al., 2012).

First, I think that more is done in this study. T. Poisot and colleagues actually
propose a method to quantify the dissimilarity between networks that takes
into account the classical beta-diversity as a component of the changes.
Second, given that this method actually has a very similar goal to the one
the authors develop in this manuscript, to me, they must carry out a meaningful
comparison of the two.




## Costs exploration

One important question related to the used of GED is how to determine
what are relevant costs in the context of communities dissimilarity?
In this manuscript, the authors explore 9 scenarios. I think this is
not enough, especially given that AMI (Adjusted Mutual Information metric) is
fairly variable among scenarios as highlighted lines 180-181:

> "It should be noted that one of the models where low cost was
assigned to interactions (values close to 0), the AMI values were higher (scenario 9; see Fig. 1)."

I believe the authors should further explore how AMI varies with costs scenarios.
For instance, they could set 5 scenarios for the cost associated with
"Flipping an interaction" (e.g. 0, .25, .5. .75 and 1) and then for each of these
scenarios they could create colored matrices that would represent how AMI
values vary with a large number of combinations of the two other costs (e.g.
from 0 to 1 with an increment of 0.01).


## Details about the method

Even though the methods section is overall clear, I have two questions that require
clarifications in the manuscript:

- Why is GED consistently higher than .6?

- Lines 148-149, the authors wrote: "We used 1,000 iterations for the calculations."
To what calculations the authors are referring to here and why 1,000 iterations are required?

Validity of the findings

To me, the findings reported here are valid and I have nothing to report in this section.
I have however suggested major modifications in the methods section that will likely
lead to new findings to report.

Additional comments

- lines 29-31:

> "The similarity between ecological communities has been one of the central topics in
ecology as community structure can impact the functioning, biodiversity, and management of
ecosystems."

I believe this sentence should be dropped as it does not add anything to the
narrative of the paper.



- line 109: "[..] from ca. The size of these networks varies from ca. 49 to 113 [...]"
I don't understand why "ca." is being used here.

- line 155: "[...] variation in cost operations on AMI [...]" what the acronym not l.163

- Errors pertaining to the formatting of references:

- lines 170-171: "(Core, 2015)" (by the way the R version should be mentioned).

- lines 70-71: "overlapping (i.e. node overlapping index; (Strona and Veech, 2015, Strona et al. 2018; Zhang et
al., 2016)".

- l.138: "(Dehmer, 2010; Ibragim format"

·

Basic reporting

In general, the manuscript is well written and clear. The introduction places the study in an appropriated context and the structure of the ms is adequate.

Experimental design

The research seems to be within the Scope of the journal and it is clearly identified how the study fills a methodological gap. The methods are clearly described, although some points could be further developed (see below).

Validity of the findings

The results appear to be robust, and the conclusions seems to be coherent with presented findings.

Additional comments

Measuring community similarity beyond species composition can provided valuable information but it is challenging. A network-based tool that considers dissimilarities in species interactions and composition is a step-on into these difficulties. In this regard, the manuscript may constitute a notable contribution. However, there were many aspects in which the study could be strengthened and/or improved.

Suggestions
A more detailed explanation of the methods would help to a better understanding. In particular, there were four points that I think would benefit from a clearer description:
1- If I understand properly the GED is based on the minimum edit path cost. Here, the cost of the edit path is the summation of costs assigned to different network transformations. If this is true, the GED should range between 0 and some number (depending on costs and number of links and nodes), and yet it seems to have an upper bound of one as if it was normalised. To a full understanding of the index, I think this normalisation should be explained. Although it might appear to be unimportant, this may help the reader to interpret the index. For instance, depending on this normalisation the index might be symmetric (i.e. equal dissimilarity when comparing network A with B than B with A) or not. Moreover, this normalization may also influence the comparison of GED values generated under different cost scenarios.

2- When describing the AMI, an explanation on how the intersection between Jaccard and GED values was calculated would help. Moreover, I think that using a commonly used correlation coefficient (e.g. Spearman’s one, if non-linear relationships) would make the results more accessible to most readers (including myself).

3- Perhaps, it would be worthy to more clearly explain the reasons behind the choice of the costs presented in Table 1, and not others.

4- An explanation on how the Jaccard index was calculated would also help. Notice that the “Jaccard index” normally refers to a similarity (intersection over union), but it seems the authors were calculating a dissimilarity index (lines 161-162).

Beyond AMI scores, it could be important a more careful interpretation of the Jaccard vs GED relationship. For instance, In Figure 1. it can be appreciated a similar shape in this relationship across scenarios. Here, it seems that at low compositional similarities (as measured by the Jaccard index), GED values present the highest variability; whereas from moderate to high Jaccard values, GED values are always high, showing little variation. The first interpretation that come to my mind is that when species composition is relatively similar, species interactions are also so and, hence, the GED “saturate”. In other words, dissimilarity due to interactions and composition may be to some extend related. Support for this interpretation come from that the relationship between GED and Jaccard appears to be stronger in the scenario 9, where GED values should be dominated by interaction editions (according to Table 1, this scenario has the lowest species/node editing costs). This interpretation might be wrong (and it is certainly contrary to the current one) but I think it exemplifies the need for a clearer and more detailed interpretation of the results.

The fact that GED values depend on arbitrary costs might be seen as a drawback. Yet, as the authors already noticed in lines 248-251, it is also one of the main advantages of the method. That is, by assigning different costs to different editions (e.g. the inclusion of nodes, or links of any type), one could “partition” the contribution of different “elements” (i.e. species, positive or negative interactions etc.) in the dissimilarity between communities. This is a very interesting idea that seems to be easy to demonstrate using the presented dataset (forgive me if I’m wrong). I think the research would largely benefit from including such demonstration and focus on it rather than on the comparison with Jaccard values. That is, I would only compare the Jaccard with GED with costs related to interaction editions equal to zero, so that I can demonstrate that the “compositional GED” is analogous to a widely used and robust index of compositional similarity. Then, I would add costs to editing interactions and compare “compositional GED” values with “compositional + interaction GED” values. This should better demonstrate the influence of changes in interactions while partially solving the issue of choosing arbitrary cost values.

A better structuration of the discussion could be also important. From the title, abstract and introduction, the ms seems to be mostly methodological and, yet, in the discussion little attention is paid to explain the advantages and disadvantages of this method compared to similar ones (e.g. the already cited Poisot et al. 2012 Ecol. Lett. 15). Providing the differences and similarities (as well as pros and cons) with other methods would help the readers (and potential users) to know when to use them. In this line, some recommendations for setting proper cost values would also help. Finally, I think that the conclusions would benefit from sticking closer to the main results of the ms and not too much to almost epistemological stuffs (i.e. what an adequate replicate of a community is?).

Minor suggestions:
- Since the ms appears to be methodological, I would first present the method and, then the dataset in the methods section.
- Figure 1 might look better if scatterplots are arranged in a 3 X 3 panel. Moreover, for an easier interpretation I would use the same values in y axes across all plots.
- In lines 148-149 it should be explained why one need several iterations (likely because GED is approached through a heuristic algorithm).
- In lines 180 to 181 it is stated that interaction editions received a low cost in scenario 9. While this is true, this scenario is the only one where species editions have lower costs than interaction editions (according to Table 1). Would this be important to the interpretation of the results?

---

## Round 0.2 · Major Revisions

Although the MS was improved, two of the referees feel further work is reauired to make the methods and text clear, and rationale behind the methods. And the third also found mistakes needing correction. Please attend to these thoroughly as I will need to send this back to the referees a third time and they would understandably be irritated is it is not well presented and their concerns well addressed.

·

Basic reporting

Dear authors,

I am glad to read the new version of your manuscript.
My previous set of comments were carefully addressed but I still have several ones that you must address before publication.

Hope this new set of comments would be helpful as you revised your manuscript.

Kevin Cazelles

Experimental design

Three comments here:

First, I think including 49 scenarios in the study is way better than 9 but I do not understand the choice of the values. Why only use 0.25 and .75 for the interactions? To me, 27 scenarios would be enough, testing 0/.5/1 for the three kind of costs.


Second, I am not satisfied by the comparison with the work of Poisot et al. (2012). I think it should be a part of the study not only "a complement". I think the AMIs values used to compare GED and the Poisot's approach should be provided for all scenarios. Alternatively, the choice of the scenarios selected must be carefully justified, which is not currently the case. I am quite surprise that the authors "found a weak relationship between BST /BWN and GED across all scenarios" and I wonder if this is only true for the selected scenarios though I hope the authors will prove me wrong.


Third, I think the authors should better explain why they have decided to use the Adjusted Mutual Information (AMI). Several questions/comments on this point:

- If I understand correctly the work of Vinh et al. (2010), the AMI is based on a contingency table based on which I(U,V) and H(U,V) are computed but I cannot figure out what is the contingency table in your approach?

- For most ecologists it may not be clear what a AMI of 0.134 (maximum value in your study) means. I understand that AHI ranges from 0 to 1 but this does not imply that 0.134 is a low value. I think this should be crystal clear as it crucial to interpret your results. One way is to create a box with a simple and well detailed example the authors could make up.

- Looks like AMI is related to a correlation coefficient, why not use a correlation coefficient instead of AMI? I have nothing against AMI but it should be better explained why you have decided to use AMI.

Validity of the findings

See point 2 above.

Additional comments

- l.95: a datasets => a dataset

- l.130: I would make explicit the three different kind of costs in this equation.

- l.145-146: implements a evolutionary => implements an evolutionary

- I think l.176-189 must come before l.166-175.

·

Basic reporting

The authors have done a great job answering my previous comments. I only have some minor suggestions/questions.

Experimental design

It is not clear to me why Poisot et al.’ beta-diversity measures are only compared with GED for a small subset of scenarios and not with all of them.

Validity of the findings

The AMI index is well explained now. However, it is still unclear to me how the mutual information is computed. That is, as far as I know measures based on mutual information are normally applied to categorical variables (e.g. cluster affiliations) and I not sure how can this be applied to quantitative variables. Perhaps it is my fault and I apologise for insisting on this, but I think this is an important detail to make clear. For instance, if one has a vector X (let’s say Jaccard values) with values 1, 2, 3, 4 and 5 and Y vector (e.g. GED) with values 1.1, 2.1, 3.1, 4.1 and 5.1; what would the AMI score be? If I understand right, both vectors are not sharing information, thus the mutual information would be 0 and so would be the AMI. Yet, both vectors are highly correlated. I think a clarification about this would help to better understand the results.

Additional comments

In lines 157-159 and 197-199 you referred to the importance of “interactions”, but it is not clear whether you meant flipping of interactions, deleting/adding interactions or both.

I found few spelling typos (but please take this carefully since I am not an English native speaker):
Line 186: “represent to” should be “represents the”
Line 247: “allow” should be “allows”
Line 291: “reduce” should be “reduced”; and “a” should be “an”.

·

Basic reporting

The authors present a methodological framework to compare how similar to ecological communities (food webs) are. To my opinion the paper is nice and the topic interesting. However, there are some parts that would need to be improved before publication.

In practical terms, one big objective of similarity measures in ecology should to allow obtaining information, in whatever means, of one ecological system given a known observed one. Take as example system stability or persistence. To illustrate this potential functionality of the framework, the paper should include experiments or solid discussion in this direction. An specific question that could be addressed is: two similar networks with respect to GED have dynamical behavior closer than two similar networks in terms of Jaccard (or $B_{WN}$, $B_{ST}$)?

After reading the paper I think it is not clear the role taxonomic identities (or the taxonomic tree) have with respect to the GED. As far as I understood, species within the different ecological systems are group according to its taxonomic similarity. In this way, two different species can be considered equivalent in the author’s representation of the ecological system. This has strong implications in the GED and in particular in its minimization. Thus, besides deciding GED costs, the main problem authors face in obtaining the minimum GED is to find which nodes of the first graph correspond to which nodes of the second graph. That is, the graph matching problem with the particularity that only taxonomic identic species can be matched. These needs to be explained in detail within the paper.

In section 2.3, line 145, authors use the word flipping a link. Later on, line 297, they explain that this refers to the change of the relation between a consumer and a resource. The mining of flipping an edge should be described already at section 2.3 otherwise the paper is not comprehensible.

I do not like the Discussion section in general and in particular the first paragraph which I consider to be misleading. Although they do not specifically say so, authors seem to indicate that the difference between the GED and Jaccard could be related to some physical or mediated factors. This is not the case for the experiments performed in the paper. The observed differences are mainly due to the fact that the different metrics measure different, probably orthogonal, aspects of the system and include different levels of information. Am I right? With respect to the rest of the discussion, I am not sure that its length cannot be reduced. Probably some parts can be merged, or moved, to the introduction to easy readability. Specifically from lines 236 to 271 which seem to be somehow describing the state of the art. This is just a proposal.

I do not like how the conclusions section is organized, to me the term summary should include the most important aspects of the paper and I probably find more interesting what is said in the second paragraph. I would consequently integrate both paragraph. Besides, I would replace item “b)” by saying “the non-relationship between the measures evidence the crucial requirement of adding structural relationships to the similarity measure” or something like that. Does it make sense?

Line 302: “variables DOES not show robust”

Experimental design

no comment

Validity of the findings

Comment on this aspect are included in the main text of the review.

Additional comments

Within the basic review text.

---

## Round 0.3 · Minor Revisions

Thank you for the acceptable revisions. Two referees make some final comments and suggestions. Please consider these and have one more careful look at the flow of the paper as this will be the last chance before you get the page proofs.

·

Basic reporting

'no comment' - see previous rouds of review.

Experimental design

'no comment'

Validity of the findings

'no comment'

Additional comments

This is my third review for the manuscript of this manuscript. The authors have answered
all my comments and I believe that the manuscript is now ready for publication in PeerJ.
I hope this publication will lead network ecologists to further investigate the potential
of the combination GED + AMI.

Below, I would like to mention a few points regarding the authors' answers to
the reviewers' comments that should not lead to any revision.


> "the reviewer 1 in the first round required to increase the number of scenarios to hundreds of parameter combinations, but now he said that 27 scenarios is enough (we performed 49 scenarios)""

I do not think that this is confusing. I just thought the 27 scenarios would have been slightly clearer, but
it is just nitpicking, I admit. 27 scenarios and above is good (again better than 9 as in the first
version of the manuscript).


> "In another example, the same reviewer suggested the use of correlation coefficient, but in a previous version another reviewer asked for removing it. In other example, reviewer 3 asked for experiments to analyze the dynamics of ecological communities with high or low similarity which is not the topic of the study at all."

My personal opinion is that when reviewers' comments are going is different directions
it often means that some parts of the manuscript lack clarity. I think if a subset of your
readership (in this case, the reviewers) are confused, it is likely that other readers
will be and so I think that instead of simply pointing out the confusion, it is better to
find out where the confusion stems from. In all fairness, such situation frequently happens when bringing news tools into a discipline.

That being said, I do think that the new version of the manuscript is clearer:
GED and AMI are well described and I think this manuscript now needs to find
its own place in the body of literature of network ecology.

·

Basic reporting

The authors have done a great job addressing all my questions. I also appreciate they provided R code to calculate AMI. This, together with their explanation, clarified how AMI was calculated. I have just one minor suggestion that might be somehow irrelevant. During the AMI calculation it is necessary to discretise the variables and how to conduct such discretisation (i.e. number of bins and discretisation method) may affect AMI scores. The authors chose default parameters of the function “discretize” from the R package “infotheo”. That is, the number of bins was equal to the cube root of the number of observations and bins have the same frequency of observations. I was playing around with these parameters and they seem to be very robust to their inferences (i.e. I got the highest AMI scores when using them, which further supports the low AMI scores they found). Thus, I think explaining that a discretisation was performed and why these parameters were chosen might potentially make clearer the AMI calculations. Nevertheless, this may turn to be somehow far from the main scope of the ms, so it is up to the authors to follow this suggestion.

Experimental design

-

Validity of the findings

-

Additional comments

-

·

Basic reporting

The authors have successfully addressed the concerns I had with the paper.

Experimental design

The authors have successfully addressed the concerns I had with the paper.

Validity of the findings

The authors have successfully addressed the concerns I had with the paper.

Additional comments

The authors have successfully addressed the concerns I had with the paper.

---

## Round 0.4 · accepted · Accept

Thank you for the final revisions. Congratulations on your contribution to the methods of community data analysis.

#